# Whole Transcriptome Analysis Identifies Platycodin D-Mediated RNA Regulatory Network in Non–Small-Cell Lung Cancer

**DOI:** 10.3390/cells11152360

**Published:** 2022-08-01

**Authors:** Shuyu Zheng, Zejuan Xie, Yanlin Xin, Wenli Lu, Hao Yang, Tianming Lu, Jun Li, Shanshan Wang, Keyu Cheng, Xi Yang, Ruogu Qi, Yongming Qiu, Yuanyuan Guo

**Affiliations:** 1Department of Biochemistry and Molecular Biology, School of Medicine & Holistic Integrative Medicine, Nanjing University of Chinese Medicine, Nanjing 210023, China; zhengshuyu@sjtu.edu.cn (S.Z.); 20210569@njucm.edu.cn (Z.X.); 20210502@njucm.edu.cn (Y.X.); 20210579@njucm.edu.cn (W.L.); 20210582@njucm.edu.cn (H.Y.); 20210512@njucm.edu.cn (T.L.); 20213075@njucm.edu.cn (J.L.); shanshanwang@njucm.edu.cn (S.W.); ckycky2000@163.com (K.C.); 2Brain Injury Center, Renji Hospital, School of Medicine, Shanghai Jiaotong University, 2000 Jiangyue Road, Shanghai 200127, China; 3Department of Neurosurgery, the Second Affiliated Hospital, Zhejiang University, Hangzhou 310009, China

**Keywords:** NSCLC, Platycodin D, RNA-seq, network pharmacology, apoptosis, cell cycle

## Abstract

Non–small-cell lung cancer (NSCLC) is one of the most fatal malignant tumors harmful to human health. Previous studies report that *Platycodin D* (PD) exhibits anti-tumor effects in multiple human cancers, including NSCLC, but the underlying mechanisms are largely unknown. Accumulating evidence indicates that non-coding RNAs (ncRNAs) participate in NSCLC disease progression, but the link between PD and the ncRNAs in NSCLC is poorly elucidated. Here, we used whole transcriptome sequencing to systematically investigate the RNAs-associated regulatory network in the PD treating NSCLC cell lines. A total of 942 significantly dysregulated RNAs were obtained. Among those, five circRNAs and six IncRNAs were rigorously selected via database and in vitro validation. In addition, the functional enrichment study of differentially expressed mRNAs, single nucleotide polymorphisms (SNPs) within PD-related mRNA structures, and the interaction between PD and mRNA-related proteins were analyzed through gene set enrichment analysis (GSEA), structural variant analysis, and molecular docking, respectively. With further in vitro validation, the results show that PD inhibits cell proliferation, arrests the cell cycle, and induces cell apoptosis through targeting BCL2-related proteins. We hope these data can provide a full concept of PD-related molecular changes, leading to a new treatment for NSCLC.

## 1. Introduction

Lung cancer is the most commonly diagnosed cancer, as well as the leading cause of cancer-related deaths worldwide [1]. According to statistics, approximately 85% of lung cancer cases are non–small-cell lung cancer (NSCLC) [2]. At present, combined therapies including surgical resection, chemotherapy, radiotherapy, and molecular target therapy are the optimal treatment for NSCLC patients [3,4]. However, the 5 year survival rate of the patients has not substantially changed, due to the lack of advancement in therapeutics [5]. Therefore, it is urgent to explore new drugs, and to identify therapeutic targets, to improve the survival probability of NSCLC patients.

*Platycodon grandiflorum* (*P. grandiflorum*) is a well-known traditional herbal medicine that is used as an expectorant for pulmonary disease and a remedy for respiratory disorders in Asia [6]. *Playtocodin D* (PD), one of the major bioactive monomers derived from *P. grandiflorum*, is reported as a potent antiproliferative and antitumorigenic agent that induces apoptosis against a panel of human cancers, including NSCLC [7,8,9]. Seo et al. report that PD could induce apoptosis and cell cycle arrest via the PI3K/Akt pathway in NSCLC [10]. Yim et al. also found the enhancement of cell death in NSCLC cells induced by *P. grandiflorum*, and the related mechanism was focused on AMPK/mTOR/AKT signal-mediated autophagy pathway [11]. However, the lack of a comprehensive pharmacological understanding of drug action mechanisms and the new insights into PD-related molecular interactions hampered the wider application of PD.

Recent advances in high-throughput next-generation sequencing (NGS) provided a variety of molecular clues about tumor initiation, maintenance, and progression in NSCLC [12]. Furthermore, transcription factors (TFs) and non-coding RNAs (ncRNAs) are two major regulators of gene expression at transcriptional and post-transcriptional levels, forming an integrated gene regulatory network by connecting TFs and ncRNAs with their interacting targets [13]. A deep investigation of this network would help to understand the molecular mechanisms of how PD exhibits anti-tumor effects on NSCLC at multiple levels. However, digging out the treasure from massive sequencing data has become a primary challenge in bioinformatics [14]. As a novel approach to analyze drug mechanisms based on the concept of “disease-gene-target-drug”, network pharmacology could systematically and comprehensively reflect the intervention mechanisms of drugs and disease [15,16,17]. The principle of network intervention is especially applicable to the treatment of tumors, as the effects of oncogenes are known to be multigenic [17]. Based on that, a comprehensive investigation of the molecular pharmacology of PD could bring new light on the treatment of NSCLC patients.

Thus, we aimed to investigate the molecular mechanisms of PD on the oncological biology of NSCLC. We performed whole transcriptome profiling (RNA sequencing, circRNA sequencing, and IncRNA sequencing) to examine the RNA regulatory networks in the PD-treated NSCLC cells. The molecular mechanism was further studied through network pharmacology and molecular docking technology. Moreover, the function of PD and its candidate molecular targets were verified through in vitro cell experiments. The data we obtained could serve as a useful resource to develop new therapeutic targets for NSCLC.

## 2. Materials and Methods

### 2.1. Cell Model and Treatment

Human lung cancer cell line A549 was purchased from the cell bank of the Chinese Academy of Sciences. They were maintained in Dulbecco’s modified Eagle medium (DMEM), supplemented with 10% fetal bovine serum (FBS) and 1% penicillin–streptomycin, at 37 °C in a humid atmosphere with 5% CO_2_. PD was purchased from Yuanye (Shanghai, China, CAS#58479-68-8) and was dissolved in sterilized water to 10 mM, aliquoted and stored at −20 °C until use. For drug treatment, the A549 cells were plated in six-well plates overnight, and treated with increasing concentrations of PD (0–20 μM) for 24 h for the following experiments.

### 2.2. RNA Extraction, Library Construction and RNA Sequencing

Three A549 cell lines treated with 8 μM PD were set as samples, and three untreated A549 cell lines were set as controls. Total RNA was extracted from each group using TRIzol Reagent (Vazyme, Nanjing, China), according to the manufacturer’s instructions. The concentration and purity of each sample was then examined with a Nanodrop 2000 spectrophotometer (Thermo Scientific, Wilmington, DE, USA), and 2% agarose gels were run to verify RNA integrity. The construction of the libraries and whole transcriptome sequencing were performed by Gikai Gene Company (Shanghai, China). Firstly, raw reads of fastq format were obtained. Adaptors and low-quality reads that contained poly-N sequences or had low Q scores were removed to obtain clean reads. The clean reads were then mapped to the reference genome using STAR (v2.5.1b). For circRNA identification, CIRI and find_circ software were used to predict circRNAs, and circRNAs that were identified by both were selected [18,19].

### 2.3. Screening of Differentially Expressed ncRNAs and Coding RNAs

The number of fragments per kilobase of exons per million fragments (FPKM) was used to estimate the expression levels of RNAs in each sample (Appendix A). Differentially expressed genes (DEGs), including coding RNAs (DE-codingRNAs) and non-coding RNAs (DE-ncRNAs) such as circRNAs, IncRNAs, and miRNAs, were identified in the control and the treatment groups using the R package (edgeR 3.14.0). Log2 fold change > 1 and adjusted *p*-value < 0.05 were set as the filter criteria for significant differential expression. The differential cluster analysis of the genes and the volcano plots of DEGs were drawn by R package (pheatmap, 1.0.12).

### 2.4. PCR and Quantitative Real-Time qPCR Validation of Candidate DE-circRNAs and DE-IncRNAs

To validate the reliability of circRNAs, we designed divergent primers encompassing back-splice junctions for five candidate DE-circRNAs (Appendix A). All circRNA primers were run on an agarose gel and were sequence-validated. The reverse transcription of circRNA was performed using HiScript II 1st Strand cDNA Synthesis kit (Vazyme, Nanjing, China). Next, real-time qPCR was performed with ChamQ Universal SYBR Green Master Mix (Vazyme, Nanjing, China) in QuantStudio Real-Time PCR Systems (ThermoFisher, Foster City, CA, USA), to validate the expression of the selected DE-circRNAs. In addition, six candidate IncRNAs were also validated by RT-qPCR. Data were quantified using the 2^−ΔΔ CT^ method, and normalized to the internal reference gene U6.

### 2.5. Weighted Gene Co-Expression Network Analysis, Gene Ontology, Pathway Enrichment Analysis, and Functional GSEA Analysis

Co-expression analysis was performed using the weighted gene co-expression network analysis (WGCNA) method via R package (WGCNA, 3.6.1). Gene ontology (GO) enrichment analysis and Kyoko Encyclopedia of Genes and Genomes (KEGG) enrichment analysis of DE-ncRNAs were performed based on their located mRNAs of DE-ncRNAs [20]. In addition, the most differentially expressed coding RNAs were subsequently analyzed by GSEA with the signal-to-noise ranking metric to determine gene sets that contained significantly lower Z scores in PD-treated cells compared to the control cells.

### 2.6. Microarray Data Analysis for the Identification of Clinical Significance of DEGs in NSCLC Patients

To evaluate the clinical importance of PD-regulated DEGs, we further used public datasets that primarily included NSCLC and healthy volunteer samples: the TCGA database (https://tcga-data.nci.nih.gov/tcga/, last accessed on 14 May 2022) and GEO datasets (GSE126533, GSE158695, GSE101684, and GSE112214, accessed on 10 May 2022). Survival analyses were performed using the Kaplan–Meier method.

### 2.7. Structurally Variation Analysis

For alterative splicing analysis, rMATS (version 4.0.2) was used to identify alternative splicing (AS) events by quantifying exon–exon junction spanning reads on annotated splice junctions in human GRCh38 assembly [21]. Differentially spliced mRNAs were defined as FDR <0.05 and an absolute inclusion level difference >10%. For visualization, an integrative genomics viewer (IGV) was used to obtain PD-regulated differential pre-mRNA AS events [22]. Variation detection was performed with GATK (version 3.1). The single nucleotide variations (SNVs) and insertions/deletions (indels) were further annotated with ANNOVAR, a publicly available suite of software for variant analysis [23].

### 2.8. Molecular Docking of PD

To predict the affinity and activity of PD to the coding-RNA-related protein targets, the molecular docking was performed by the LibDock module embedded in Discovery Studio 2019 (BIOVIA, San Diego, CA, USA). The protein structures of the BID, RPDIA, and Cyclin E were retrieved from PDB database, and the chemical structure data of PD (SDF file) was obtained from PubChem (https://pubchem.ncbi.nlm.nih.gov/) (accessed on 5 March 2022). The results were evaluated using the LibDock score, with a higher LibDock score indicating a higher activity of the small molecule (such as PD) binding to the target protein.

### 2.9. Cell Apoptosis Assay and Cell Cycle Analysis

The cell apoptosis assay and cycle analysis were performed with a flow cytometer (BD Biosciences, San Jose, CA, USA). Propidium iodide (PI; Vazyme, Nanjing, China) was used for cell cycle analysis, while a staining kit containing Annexin V-fluorescein isothiocyanate (Annexin V-FITC) and PI (Vazyme, Nanjing, China) was used to assess cell apoptosis. A549 cells (2 × 10^5^) were seeded in six-well plates for 24 h, and then incubated with increasing concentrations of PD (0–20 μM) for 24 h. For cell cycle analysis, the cells were washed with cold phosphate-buffered saline (PBS) and then fixed with chilled 75% ethanol overnight at −20 °C. After being centrifuged at 1000 rpm for 5 min, the cells were washed with chilled PBS, and were stained with PI for 30 min at 37 °C for flow cytometer analysis. For the apoptosis experiment, after treatment with different concentrations of PD, the cells were harvested with EDTA-free trypsin and centrifuged at 1000 rpm for 5 min, followed by washing twice with PBS. Then, these cells were resuspended in 200μL 1× binding buffer, and then incubated with PI and Annexin V-FITC for 15 min at room temperature in the dark, according to the manufacturer’s instructions (Vazyme, Nanjing, China). Flow cytometry was performed, and a total of 10,000 events were collected per sample. Further, the apoptosis and cell cycle profiles were analyzed by Kaluza Analysis software. Each experiment was performed in triplicate.

### 2.10. Statistical Analysis

Data are expressed as the mean ± SD of triplicates. Each experiment was repeated at least 3 times. Student’s *t*-test or ANOVA were used to evaluate differences using GraphPad Prism software (GraphPad Prism 9, San Diego, CA, USA). * *p* < 0.05; ** *p* < 0.01; *** *p* < 0.001; **** *p* < 0.0001.

## 3. Results

### 3.1. Overview of RNA-Seq Data

In total, 614,837,388 raw reads were generated: 101,082,150, 105,049,018, and 106,651,964 for PD-treated A549 cell replicates, along with 92,381,028, 104,249,030, and 105,424,198 for the controls. After removing low-quality reads and those containing adaptors and poly-N, 595,768,762 clean reads remain.

### 3.2. Identification of Differentially Expressed Genes

We identified 26 significantly dysregulated circRNA transcripts between the two groups (PD-treated vs. the control A549 cells). Among those, 10 circRNAs are significantly up-regulated and 16 circRNAs are down-regulated in PD-treated A549 cells compared to the control cells (fold change > 2 and *p* < 0.05) on volcano plots (Figure 1A). In addition, for IncRNA transcripts, we found 380 dysregulated transcripts between the two groups, with 211 and 169 being up-regulated and down-regulated, respectively, in PD-treated groups (Figure 1B). Moreover, a total of 536 mRNA transcripts are identified as significantly differentially expressed mRNAs based on FPKMs value: 300 mRNA transcripts are up-regulated and 236 of those are down-regulated (Figure 1C). Additionally, hierarchical cluster analyses of the differentially expressed circRNAs, IncRNAs, and mRNAs suggest good consistency among the three replicates (Figure 1D and Appendix A).

### 3.3. Enrichment and Functional Analysis of DE-ncRNAs-Related Target Genes

To better understand the biological function of the DE-ncRNAs, we then conducted GO and KEGG enrichment and functional analysis of DE-ncRNAs-related host/target genes. Among all circRNAs, 1320 circRNAs are mapped to 1281 host genes; for DE-IncRNAs, 1,048,575 target genes are predicted. The top highly enriched GO terms of biological process (BP), cellular component (CC), and molecular function (MF) are shown in Figure 2A. The results show that the host/target genes are involved in various signaling pathways, participating in energy metabolism, cell junction organization, cell proliferation, and signal transduction (Figure 2B,C).

### 3.4. qPCR Validation of DE-ncRNAs

Based on the differential expression analysis in our sequencing data, hsa_circRNA_0001326, hsa_circRNA_0001946, hsa_circRNA_0005870, hsa_circRNA_0006702, and hsa_circRNA_0006990 were selected for further analysis by qPCR. By searching the Genome Reference Consortium Human Build 37 (GRCh37/hg19), we found that hsa_circRNA_0001326 is situated on chromosome 03 (chr03 (+strand): 111,632,165–111,639,266), residing within the protein PHLDB2; hsa_circRNA_0001946 is situated on chromosome X (chrX (+strand): 139,865,339–139,866,824), residing within the protein CDR1; hsa_circRNA_0005870 is situated on chromosome 03 (chr03 (−strand): 47,079,155–47,088,111), residing within the protein SETD2; hsa_circRNA_0006702 is situated on chromosome 09 (chr09 (−strand): 95,018,961–95,032,265), residing within the protein IARS; and hsa_circRNA_0006990 is situated on chromosome 18 (chr18 (+strand): 9,931,806–9,937,063), residing within the protein VAPA. To rigorously evaluate the expression of aforementioned circRNAs in the cells, we firstly amplificated five circRNAs with divergent primers (Appendix A), followed by Sanger sequencing across the back-splicing junction sites (Figure 3A,B, Appendix A). Then, we tested the expression level of these five circRNAs after treating A549 cells with PD by RT-qPCR; the results reveal that, compared with the control, hsa_circ_0001326 is significantly up-regulated in PD-treated cells, while hsa_circ_0001946 and hsa_circ_0006702 are significantly down-regulated in PD-treated cells (Figure 3C). For selected lncRNAs, we found a dramatic increase in NEURL3 in PD-treated A549 cells through RT-qPCR (Appendix A).

### 3.5. GSEA Enrichment Analysis

To further identify the functional roles of DE-coding RNAs, we used GSEA enrichment analysis to distinguish the key functional process between the A549 and PD-treated A549 cell groups. As shown in Figure 4, the cell-cycle-related pathways are markedly enriched in the A549 cells, and are significantly down-regulated in PD-treated A549 cells, including cell cycle checkpoints, formation of the β-catenin, TCF-transactivating complex, G2/M DNA damage checkpoint, homology directed repair, mitotic prophase, and mitotic spindle checkpoint.

### 3.6. Structural Variant Analysis

We also analyzed the structural variants between A549 cells and PD-treated A549 cells based on microarray sequencing. The significantly associated SNPs were filtered and identified within a 1 Mb window size (500 Kb upstream and downstream), using the human reference genome assembly (GRCh37/hg19, Figure 5A). Interestingly, we found in the PD-treated A549 cells sequencing data the SNPs related to cell cycle that are predominantly identified, including RPRD1A rs4799407 A > C, BCL2 rs1541295 C > T, BID rs181392 T > C, and HUS1 rs2708902 G > A (Figure 5B). In addition, the analysis of alternative splicing events dysregulated in PD-treated A549 cells sequencing shows a high level of affected exon usage using rMATS software (including retained introns, mutually exclusive exons, alternative 3′and 3′ splice sites, alternative 5′and 3′ splice sites, and skipped exons, Figure 6A). As predicted by the rMATS analysis, compared to the control, A549 cells treated by PD exhibit a significant change in alternative splicing of BCL2 (Figure 6B).

### 3.7. Assessment in TCGA and GEO Cohorts

To further explore the clinical significance of the aforementioned differentially expressed genes, we firstly conducted differential expression analysis in TCGA and GEO cohorts, and analyzed the expression of hsa_circRNA_0001326, hsa_circRNA_0001946, hsa_circRNA_0005870, hsa_circRNA_0006702, and hsa_circRNA_0006990 in NSCLC patients. The data show that, compared to the healthy volunteers, hsa_circRNA_0001326 and hsa_circRNA_0001946 are significantly down-regulated in the serum of NSCLC patients, while hsa_circRNA_0005870, hsa_circRNA_0006702, and hsa_circRNA_0006990 are significantly up-regulated in the serum of NSCLC patients (*p* < 0.05, Figure 7). Next, the structural information about these five circRNAs is demonstrated using the online web tool (http://geneyun.net/CSCD2/) (accessed on 7 March 2022). We also used CSCD to forecast the miRNAs response elements (MREs). As a consequence, a total of 22 miRNAs are discovered at the downstream of hsa_circRNA_0001326, 27 miRNAs at the downstream of hsa_circRNA_0001946, 37 miRNAs at the downstream of hsa_circRNA_0006702, and 27 miRNAs at the downstream of hsa_circRNA_0006990 (Figure 8A). Further, we conducted weighted gene co-expression network analysis based on the expression matrix of DE-circRNAs selected from TCGA and GEO cohorts. The results show the key regulatory roles of hsa_circRNA_0001326, hsa_circRNA_0001946, hsa_circRNA_0005870, hsa_circRNA_0006702, and hsa_circRNA_0006990 in the co-expression network (Figure 8B).

For IncRNAs, we also conducted the differential expression analysis in the TCGA database, and 127 IncRNAs were sorted in ascending order of P.adj value (Figure 9A,B). Among those, C6orf223, SNAI3-AS1, NEURL3, IGFL2-AS1, LINC02243, and LINC02643 are ranked among the top 20 DE-IncRNAs. In addition, survival analysis by KM plotter (http://kmplot.com/) (accessed on 04 March 2022) shows that the low expression levels of C6orf223, SNAI3-AS1, NEURL3, IGFL2-AS1, and LINC02243 are associated with short survival in NSCLC patients, and that high expression levels of LINC02643 are related to poor prognosis in NSCLC patients (Figure 9C).

### 3.8. Molecular Docking

To further explore the mechanism of interaction between PD and DEGs, we then constructed a molecular docking model of PD and the DEGs. We found that BID (PDB ID: 5C3F, LibDock score: 151.849), RPRD1A (PDB ID: 4JXT, LibDock score: 134.107), and Cyclin E (PDB ID: 1W98, LibDock score: 129.988) exhibit the stable binding ability to the PD small molecule, and the residues of PD interact with hydrogen bonds in the binding sites (Figure 10).

### 3.9. Evaluation the Effect of PD on Cell Proliferation, Cell Cycle, and Apoptosis

Based on the aforementioned analysis, we further evaluated the effects of PD on A549 cell proliferation, cell cycle, and apoptosis. As shown in Figure 11A, PD activates apoptosis in A549 cells in a dose-dependent manner. The percentage of cells undergoing apoptotic cell death increases from 1.24% in the control culture to 58.31% in A549 cells following the treatment of 15 μM PD for 24 h. In addition, the various concentrations of PD treatment result in the increased accumulation of cells in the G2/M phase, and a corresponding decrease in the G0/G1 and S phases, suggesting the block of G2 progression (Figure 11B). Furthermore, PD inhibits cell proliferation of both 2D and 3D culture A549 cells, consistent with the visualization of live/dead colonies from the Calcein AM/PI staining (Figure 11C, Appendix A).

## 4. Discussion

In the past decade, a growing number of therapies were identified to treat NSCLC patients, but only a few treatments exhibited the effect of increasing the survival rates and improving the quality of the patients’ life. This situation is attributed to the complex regulatory network of NSCLC pathogenesis, relating to multiple genes and pathways [24,25]. Therefore, targeting multiple genes is likely to be the future direction of the treatment of NSCLC [23]. Comfortingly, the unique advantages of traditional Chinese medicine (TCM), such as multitarget interventions, became an indispensable systemic approach for patients [26,27]. The present study provided new insights into the molecular mechanisms of PD using whole transcriptome sequencing and related experiments, which represent a novel therapeutic direction for NSCLC treatment.

In recent years, the pivotal role of the ncRNAs in the tumorigenesis and progression of NSCLC was demonstrated. In particular, as a category of important heterologous ncRNAs with a length greater than 200 nt [28], IncRNAs play a critical role in the occurrence and development of NSCLC through the following mechanisms [29]. LncRNAs trigger the EMT process, reprogram the chromatin state through PRC2/LSD1, and interact with tumor suppressor miRNAs, which act as sponges for miRNAs to interfere with miRNA–target gene interaction [30]. In our study, we discovered and validated six differentially expressed IncRNAs in NSCLC cells under PD treatment. Among those, NEURL3 and IGFL2-AS1 are the most stably regulated IncRNAs in a PD dose-dependent manner.

IncRNA NEURL3 is reported to be involved in the hematopoietic stem cell (HSC) development. Combing bioinformatics analysis and in vivo functional evaluation, Hou et al. found that the expression profile of NEURL3 varies with the development of HSC-primed human endothelial cells, and NEURL3-EGFP labeling was used as a functional tool to monitoring this process [31]. However, the role of NEURL3 in human cancer has not yet been explored. For the first time, we found that NEURL3 is significantly up-regulated in PD-treated NSCLCs, and its low expression is related to poor prognosis. Besides, previous studies demonstrate that IGFL2-AS1 promotes proliferation, migration, and invasion in epithelial cancers, such as colon cancer, breast cancer, and tongue squamous cell carcinoma [32,33,34]. Wang et al. report that in basal-like breast cancer (BLBC) cells, IncRNA IGFL2-AS1 is the downstream target gene of KLF5, which is an oncogenic transcription factor in BLBCs. Through up-regulating its neighboring gene IGFL1, IGFL2-AS1 c mediates the pro-proliferation and pro-survival functions of KLF5 in BLBCs [34]. In our study, we found that IGFL2-AS1 is up-regulated in PD-treated NSCLCs, and the K–M analysis also shows that the low expression of IGFL2-AS1 is related to the poor prognosis of NSCLC, suggesting a protective role of IGFL2-AS1 in NSCLC. This opposite effect possibly emanated from the differences in the microenvironment between the NSCLC and other epithelial cancers.

Another category of ncRNAs with distinguishable significance are circRNAs, which have covalently closed-loop structures, endowing them with a higher stabilization [35]. Thus, using circRNAs as diagnostic or predictive biomarkers for NSCLC recently gained increased attention [36]. Zhang et al. detected the expression of hsa_circ_0001946 in the freshly frozen tumor tissues and adjacent tissues of 284 NSCLC patients [37]. They found that the expression of hsa_circ_0001946 in adjacent tissues is over three-fold that in tumor tissues, and the higher expression of hsa_circ_0001946 is related to less lymph node metastasis and decreased TNM stage in NSCLC patients, suggesting that hsa_circ_0001946 could be used as a prognostic biomarker in NSCLC patients [38]. In addition, Fan et al. report that hsa_circ_0001946 detected in the blood could also be used as a prognostic biomarker in esophageal squamous cell cancer patients, with the AUC, sensitivity, and specificity of 0.894, 92, and 80%, respectively [39]. Consistently, we found a remarkable elevated expression of hsa_circ_0001946 in PD-treated cell lines, compared with that in the control, suggesting a salvage role of hsa_circ_0001946 in NSCLC. However, some controversies still exist regarding this subject. Yao et al. found hsa_circ_0001946 promotes cell growth in lung adenocarcinoma by regulating the miR-135a-5p/SIRT1 axis and activating the Wnt/β-catenin signaling pathway [39]. Therefore, the mechanism of hsa_circ_0001946 needs further investigation.

The inhibition of cancer cell proliferation, followed by the induction of cell apoptosis is targeted in chemotherapeutic strategies for the treatment of NSCLC. The anti-apoptotic apoptosis-related B cell lymphoma-2 (BCL2) family proteins are emerging as important targets for NSCLC treatment [40,41]. For example, the BCL2 inhibitor drug navitoclax was used in clinical trials, and shows efficacy in combination with other targeted therapy drugs, such as EGFR inhibitors, in EGFR-mutation-positive NSCLC [42]. Unfortunately, it also exhibits dose-limiting toxicities including neutropenia and thrombocytopenia [43]. In our study, we demonstrate that PD could be a promising strategy for adjuvant treatment of NSCLC. Using network pharmacology, we found interactions between PD, cell-cycle-related proteins and apoptosis-related proteins, such as BCL2 and BH3-interacting domain death agonist (BID). Further, the in vitro study confirms the inhibitory effects of PD on carcinoma cell proliferation and cell cycle progression, as well as the promotion effects of PD on cell apoptosis. Through in silico analysis, we find a docked pose of BID after molecular docking with PD. Structural variant analysis further finds the SNP site located in BID (rs181392 T > C) in PD-treated cells. If further proven in an in vivo environment, this would translate to a blockade of the BCL2-related proteins, particularly in BID. From this result, it could be extrapolated that PD could possibly be a potent apoptosis-inducer in the NSCLC treatment. The limitations of this study include the small sample size, and the heterogeneity of the study group, which may lead to statistical bias. Moreover, the interaction between PD and BCL2 family proteins needs further investigation.

In summary, we elucidated the whole transcriptome profile of PD-treated NSCLC cells. Combing RNA-seq analysis and database validation, five circRNAs and six IncRNAs were selected as PD-responsive ncRNAs, with clinical significance. In addition, PD inhibits cell proliferation and the cell cycle, and induces cell apoptosis through targeting BCL2-related proteins. This study, for the first time, reveals the comprehensive molecular mechanism of PD intervention, which could provide new insights to explore the regulatory modules underlying the effects of TCM on diseases.

## Figures and Tables

**Figure 1 cells-11-02360-f001:**
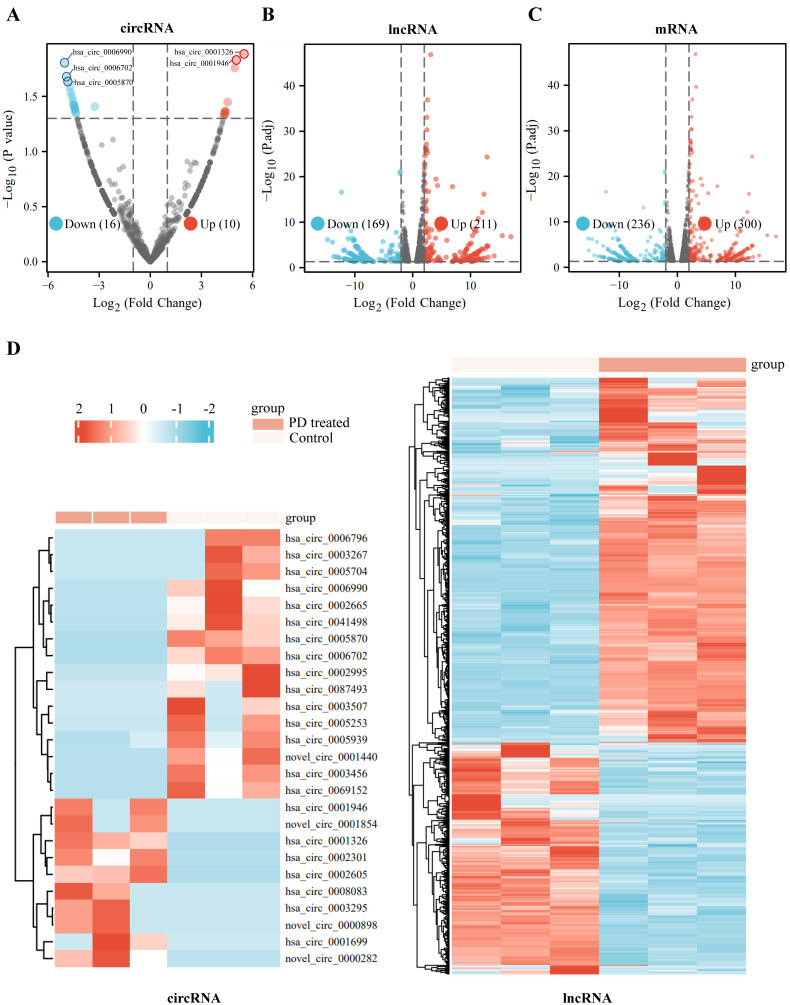
Expression profiles of differentially expressed RNAs in the PD-treated cells and the controls. (**A**–**C**) Volcano plots of differentially expressed circRNAs (**A**), differentially expressed lncRNAs (**B**), and differentially expressed mRNAs (**C**) between the PD-treated A549 cells group and the controls. X axis: log2 ratio of circRNA expression levels (**A**), lncRNA expression levels (**B**), and mRNA expression levels (**C**) between the two groups. Y axis: false discovery rate values (−log10 transformed) of *p* values. (**D**) Heatmaps of differentially expressed circRNAs and differentially expressed lncRNAs between the PD-treated A549 cells and the controls. Red indicates up-regulation, and blue indicates down-regulation.

**Figure 2 cells-11-02360-f002:**
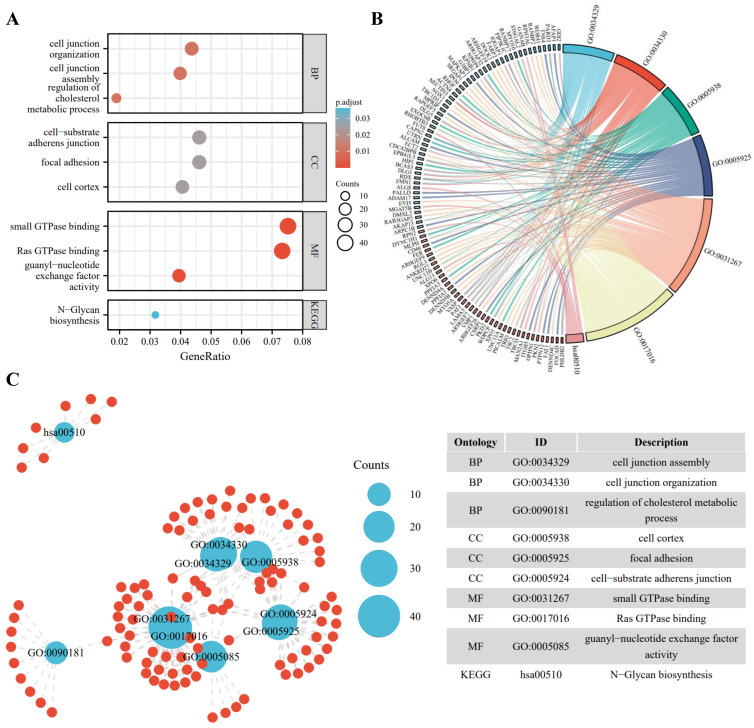
Gene ontology (GO) enrichment annotations and Kyoto Encyclopedia of Genes and Genomes (KEGG) analysis of DE-ncRNAs-related pathways. (**A**) Functional enrichment of DE-ncRNAs in the PD-treated A549 cells vs. the control. Top 3 enriched biological process (BP), cellular component (CC), molecular function (MF), and the most enriched KEGG pathways are listed. (**B**) Chord diagram of DE-ncRNAs-associated genes and GO pathways. Significantly enriched GO pathways featured *p* values < 0.05. Each line represents a gene, and the number of lines indicates the genes enriched. (**C**) Enrichment map displays the enriched DE-ncRNAs-associated GO and KEGG pathways in the PD-treated A549 cells vs. the control. Blue nodes represent items and the red nodes represent molecules. Node size represents the gene-set size and the dot lines represent the connection between the corresponding items and the molecules.

**Figure 3 cells-11-02360-f003:**
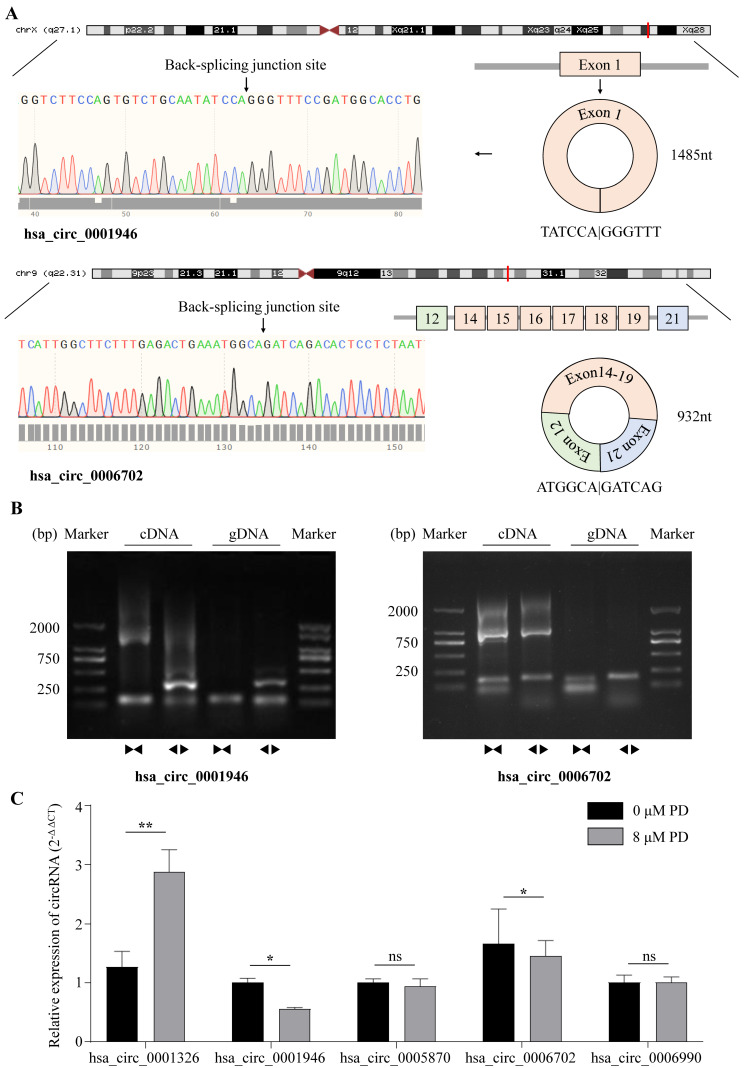
Characterization of DE-circRNAs. (**A**) Genomic location and splicing pattern of hsa_circ_0001946 and hsa_circ_0006702. The back-splicing site was verified by Sanger sequencing. (**B**) Identification of hsa_circ_0001946 and hsa_circ_0006702 qPCR amplification products. DNA gel electrophoresis shows divergent primers amplify hsa_circ_0001946 and hsa_circ_0006702 in cDNA but not in gDNA, convergent primers amplify both linear hsa_circ_0001946 and hsa_circ_0006702. (**C**) The relative expression of selected 5 circRNAs (hsa_circ_0001326, hsa_circ_0001946, hsa_circ_0005870, hsa_circ_0006702, and hsa_circ_0006990) in A549 cells treated with 8 μM PD for 48 h is determined by qRT-PCR. Β-actin and GAPDH were used as internal control. * *p* < 0.05; ** *p* < 0.01; ns stands for not statistically significant.

**Figure 4 cells-11-02360-f004:**
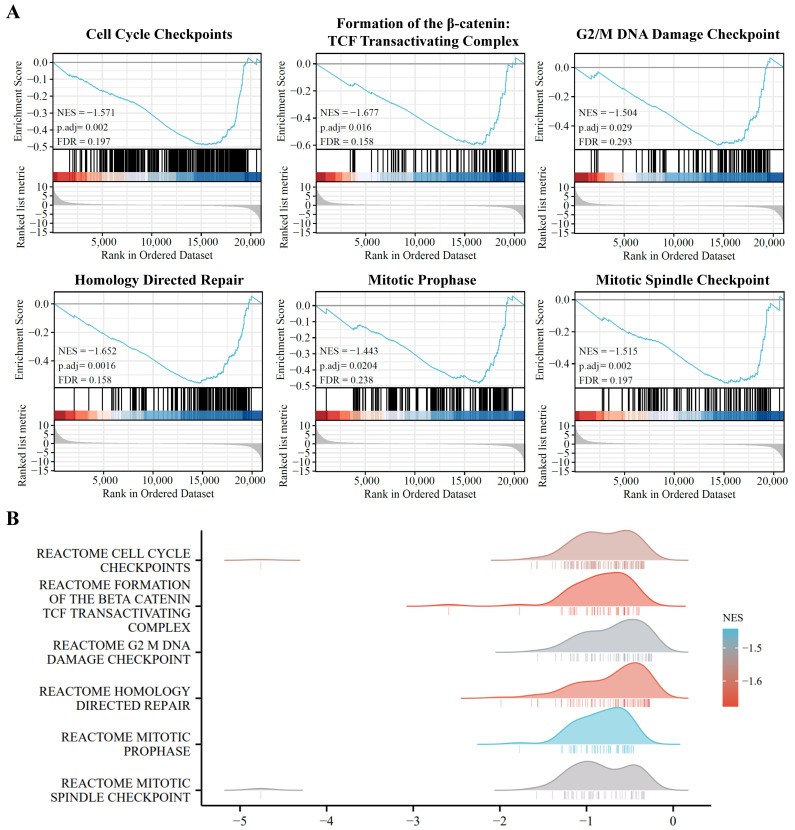
Pathway enrichment analysis of the DE-mRNAs. (**A**) Gene set enrichment analysis (GSEA) displays the top 6 enriched pathways in DE-mRNAs. (**B**) Ridge plots with normalized enrichment score (NES) display these most enriched pathways in DE-mRNAs in GSEA.

**Figure 5 cells-11-02360-f005:**
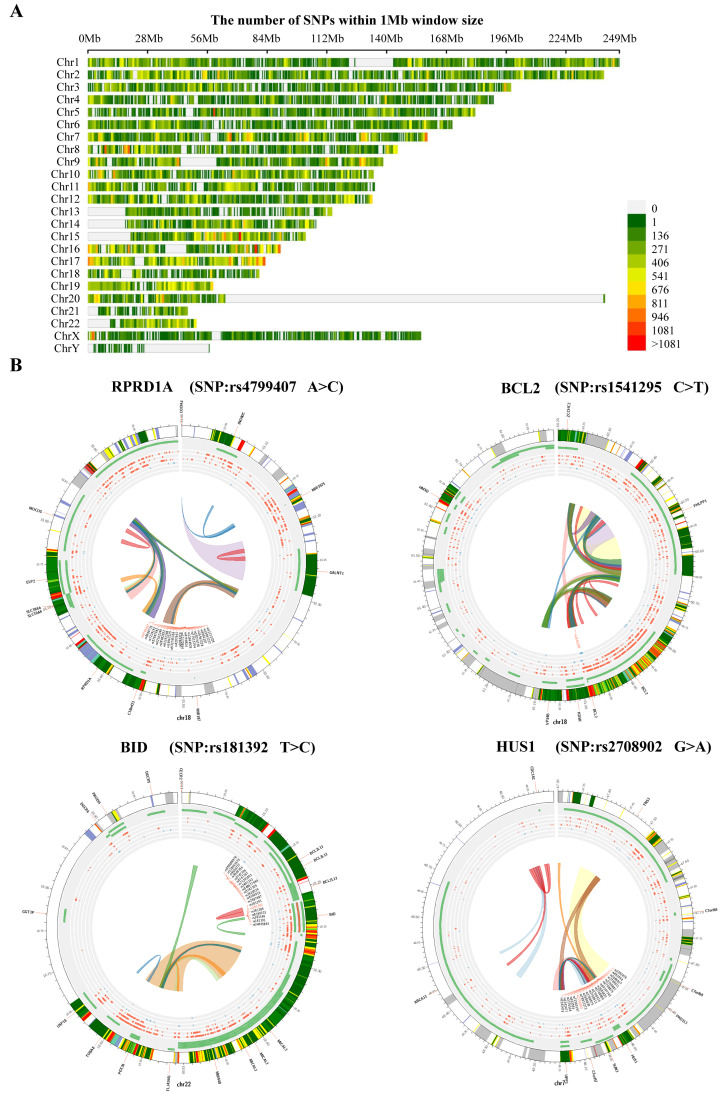
The analysis of DE-mRNAs-associated SNPs. (**A**) Chromosome distribution of SNPs within 1 Mb window size. (**B**) The top 4 genetic polymorphisms (rs4799407, rs1541295, rs181392, and rs2708902) are shown in the Circos plots. From outer to inner, the circles represent chromatin states, annotated genes (green), histone modification set (red), transcription factor set (blue), current variant and associated variants, and 3D chromatin interactions.

**Figure 6 cells-11-02360-f006:**
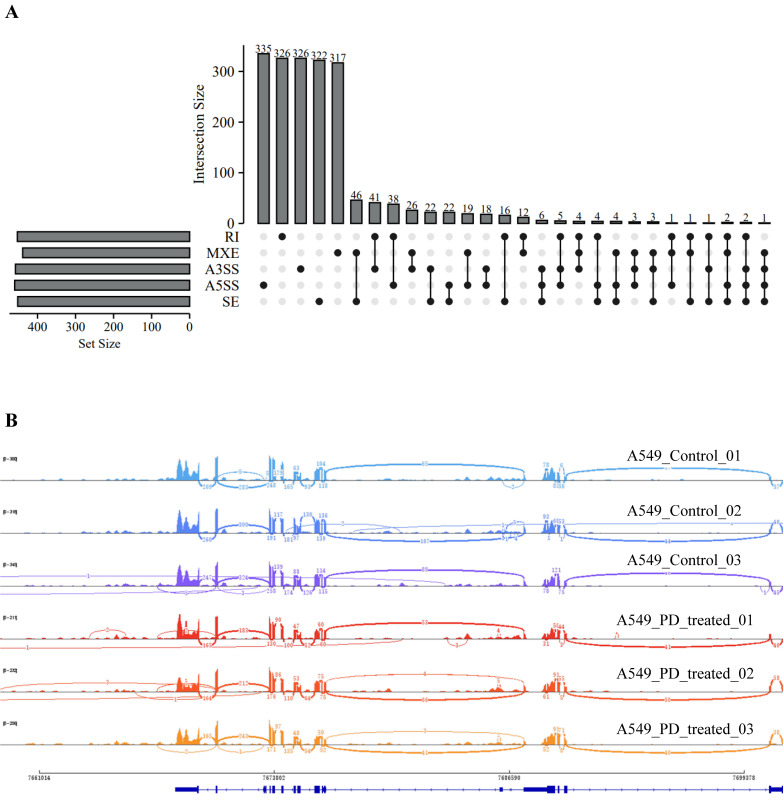
Analysis of alternative splicing events in DE-mRNAs between the PD-treated A549 cells and the control. (**A**) An UpSet diagram exhibits five types of alternative splicing events for DE-mRNAs. (**B**) Sashimi plots for the alternatively spliced BCL2 gene in the two groups.

**Figure 7 cells-11-02360-f007:**
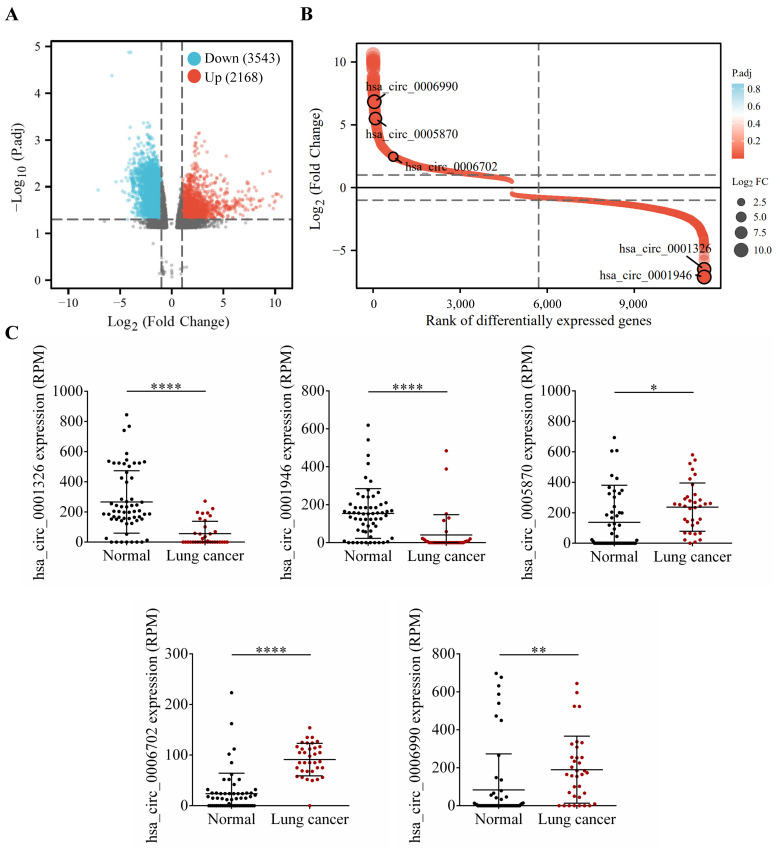
Identification and analysis of circRNAs from TCGA and GEO database. (**A**) Volcano plot shows the profile of differentially expressed circRNAs between the healthy volunteers and NSCLC patients from TCGA and GEO databases. (**B**) has_circ_0006990, hsa_circ_0005870, hsa_circ_0006702, hsa_circ_0001326, and hsa_circ_0001946 are screened by robust rank aggregation. (**C**) The relative expression of these DE-circRNAs is further verified. * *p* < 0.05; ** *p* < 0.01; **** *p* < 0.0001.

**Figure 8 cells-11-02360-f008:**
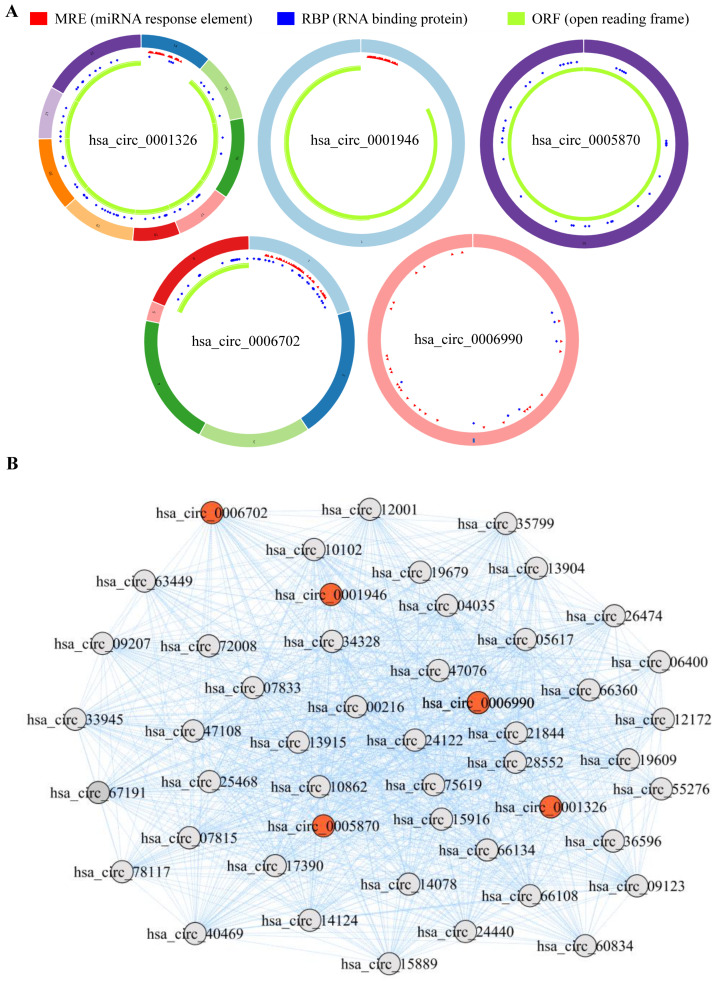
The interaction between the DE-circRNAs and other molecules. (**A**) The fundamental structures of 5 DE-circRNAs are predicted by CSCD. (**B**) RNA regulatory network based on weighted gene co-expression network analysis exhibits the key regulatory functions of 5 DE-circRNAs.

**Figure 9 cells-11-02360-f009:**
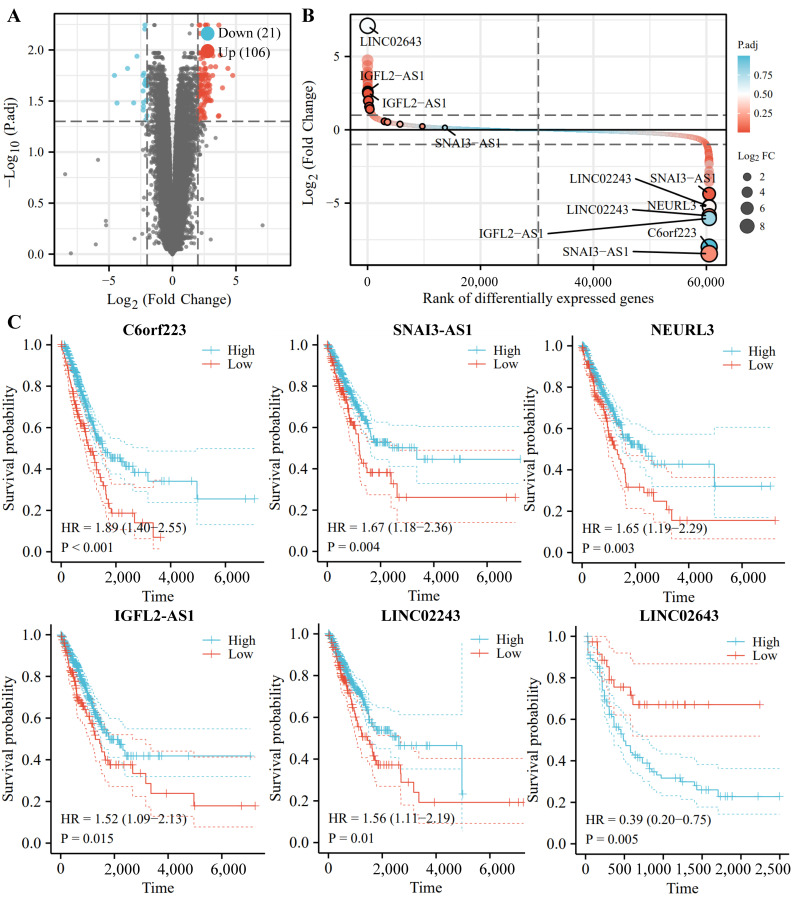
Identification and analysis of lncRNAs from TCGA and GEO database. (**A**) Volcano plot shows the profile of differentially expressed lncRNAs between the healthy volunteers and NSCLC patients from TCGA and GEO databases. (**B**) LINC02643, LINC02243, IGFL2-AS1, NEURL3, SNAI3-AS1, and C6or223 are screened by robust rank aggregation. (**C**) The KM plots of DE-lncRNAs.

**Figure 10 cells-11-02360-f010:**
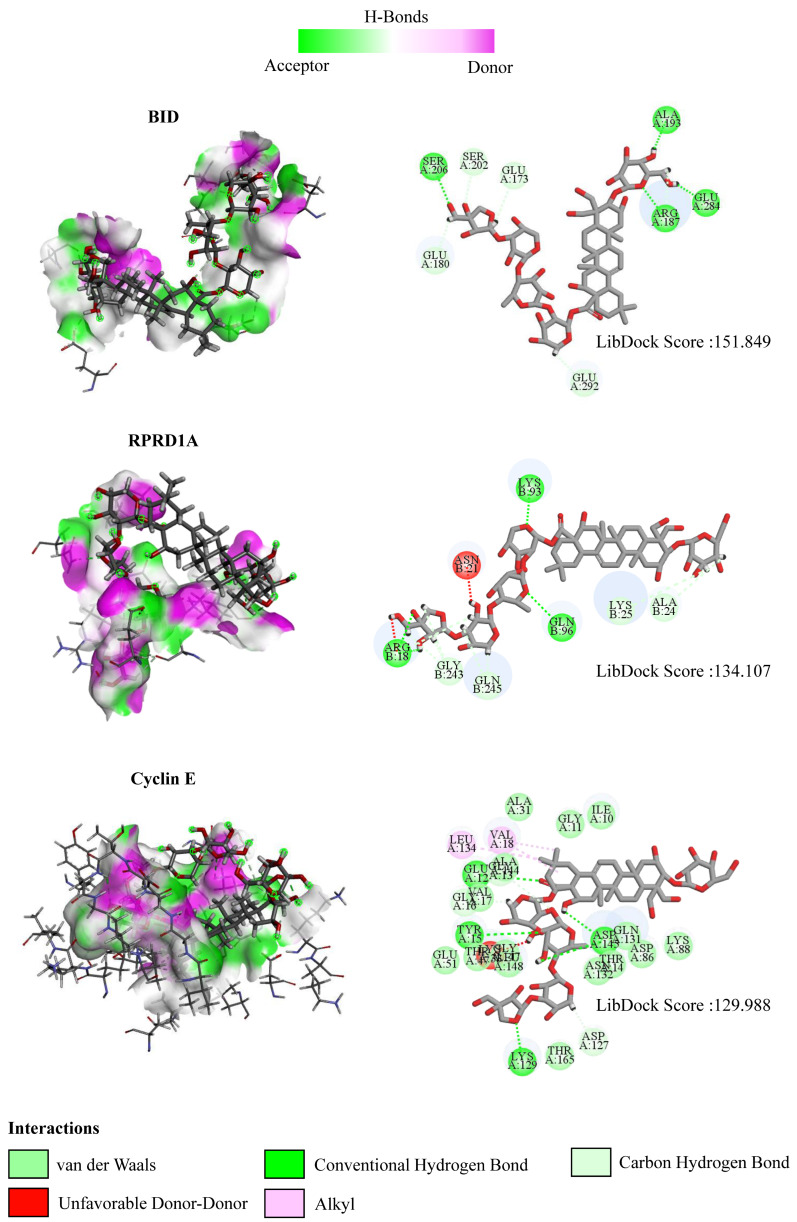
The interactions between PD and target proteins. The interactions between PD and BID, RPRD1A, and Cyclin E proteins are shown. In H-bonds, pink indicates the donor and green indicates the receptor. In the interactions, green indicates van der Waals interactions, dark green indicates conventional hydrogen bonds, light green indicates carbon–hydrogen bonds, dark red indicates unfavorable donor–donor interactions, and pink indicates alkyl interactions. The LibDock scores are calculated by Discovery Studio 2019.

**Figure 11 cells-11-02360-f011:**
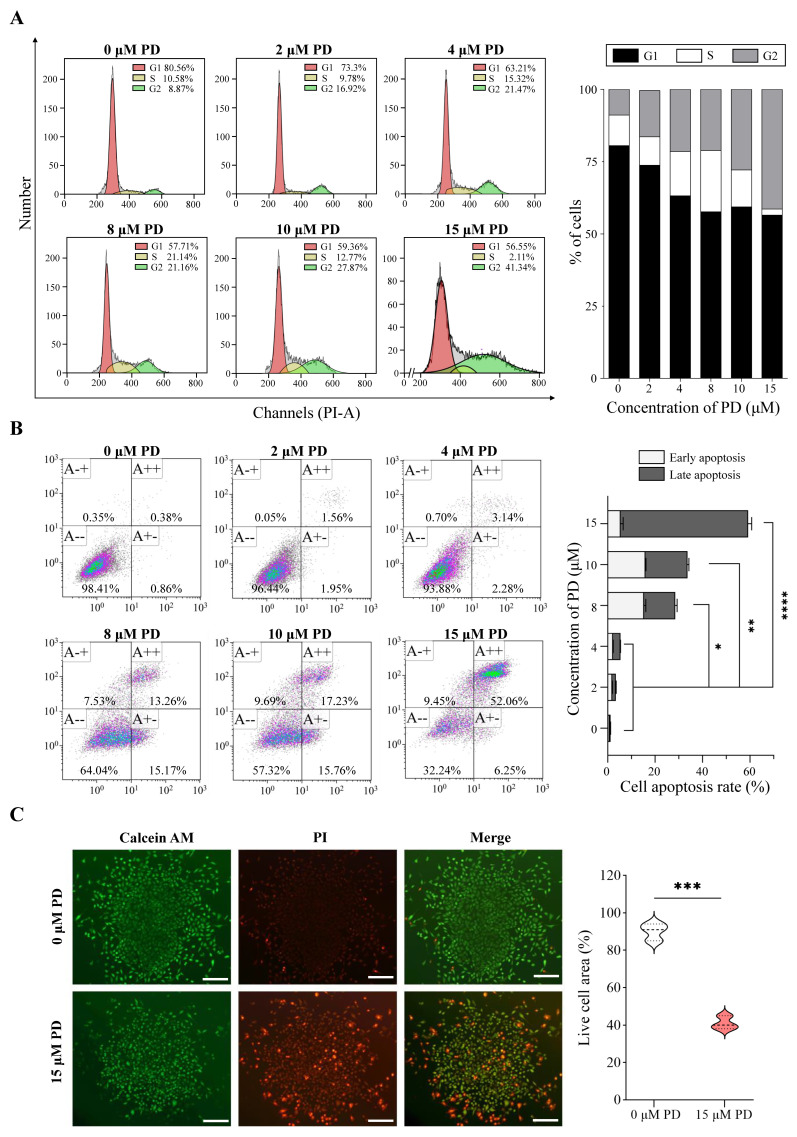
PD suppresses cell proliferation and colony formation, and induces cell apoptosis in A549 Cells. Flow cytometry results show the cell cycle changes (**A**) and apoptosis induction (**B**) in PD-treated A549 cells. (**C**) Colony formation. Live cells are stained by Calcein–AM (green color), and dead cells are stained by PI (red color). Scale bar: 100 μm * *p* < 0.05; ** *p* < 0.01; *** *p* < 0.001; **** *p* < 0.0001.

## Data Availability

Please contact the corresponding author for all data requests.

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
