# Peer review of "Whole Transcriptome Analysis Identifies Platycodin D-Mediated RNA Regulatory Network in Non–Small-Cell Lung Cancer"

_cells, 2022, doi:10.3390/cells11152360_

Round 1

Reviewer 1 Report

The manuscript titled "Whole Transcriptome Analysis Identifies Platycodin D Mediated RNA Regulatory Network in Non-small Cell Lung Cancer" by Shuyu Zheng et al., is a very well-designed research work exploring the role of ncRNAs in NSCLC pathogenesis and therapeutic opportunities. In particular, the authors demonstrated that starting from the selected ncRNAs whose expression highly depended on PD treatment, 5 circRNAs, and 6 IncRNAs showed an encouraging clinical significance. Furthermore, the methods adequately fit with the aim of the study, and overall the work is well supported by a careful reading of the literature in the field.

Author Response

Thank you for reading our paper carefully and giving the positive comments. 

Reviewer 2 Report

1.    [method section] Please include read preprocessing and mapping tools used in this study, especially how circular RNAs were identified.
2.    In the method section, the authors described TPM for the identification of differentially expressed genes. Whereas, in results section 3.2, FPKMs were used instead. Please clarify.
3.    Please include a sample supplementary table with replicates, number of reads, transcripts etc. identified after mapping.
4.    Line 185: instead of “remarkably” up-regulated, please consider using “highly” or “significantly”. Also, please include the names of highlighted transcripts in volcano plots (Figure 1A).
5.    Line 190-191. The term “significantly changed”, is it different from differential expression?
6.    Figure 1E is missing. Please recheck the labels and legends.
7.    In Figure 2A legends please replace DEG with DEncRNA.
8.    Section 3.4 (Line 226): “Based on the differential expression analysis in our initial microarray data “. Which microarray data authors trying to mention here. I believe the dataset in the previous section is based on the sequencing platform.
9.    Please include WGCNA details in the method section.
10.  Please include a limitation section and mention the possible variables that can change the outcome, e.g. sample size.

Author Response

Thank you very much for giving us the opportunity to revise the manuscript. The comments and suggestions raised here were carefully analyzed and considered by all of the authors, and we tried our best to revise our manuscript according to the comments. The detailed, point-by-point responses to the reviewer comments are given below. Please see the attachment for more details.  

Point 1: [method section] Please include read preprocessing and mapping tools used in this study, especially how circular RNAs were identified.

Response 1: Thank you for your constructive comment. We have revised the manuscript in line 103-108 “Raw reads of fastq format were firstly obtained. Adaptors and low-quality reads that contained poly-N sequences or had low Q scores were removed to obtain clean reads. Then the clean reads were mapped to the reference genome using STAR (v2.5.1b). For circRNA identification, CIRI and find_circ software were used to predict circRNAs, and circRNAs which were identified by both were selected [18,19]”. And we also add two more references in this statement.

[18]. Memczak, S., Jens, M., Elefsinioti, A., Torti, F., Krueger, J., Rybak, A., Maier, L., Mackowiak, S. D., Gregersen, L. H., Munschauer, M., Loewer, A., Ziebold, U., Landthaler, M., Kocks, C., le Noble, F., & Rajewsky, N. (2013). Circular RNAs are a large class of animal RNAs with regulatory potency. Nature, 495(7441), 333–338.

[19]. Gao, Y., Wang, J., & Zhao, F. (2015). CIRI: an efficient and unbiased algorithm for de novo circular RNA identification. Genome biology, 16(1), 4.

Point 2:  In the method section, the authors described TPM for the identification of differentially expressed genes. Whereas, in results section 3.2, FPKMs were used instead. Please clarify.

Response 2: Thank you for your careful check. We are sorry for making this misguided statement. For whole transcriptome sequencing data, FPKM was used to estimate the expression levels of RNAs in each sample. We have revised the corresponding statement in line 110-111 in the manuscript.

Point 3: Please include a sample supplementary table with replicates, number of reads, transcripts etc. identified after mapping.

Response 3: We are very grateful to your comment. We have added relevant materials as the reviewer suggested in the supplementary file. 

Point 4: Line 185: instead of “remarkably” up-regulated, please consider using “highly” or “significantly”. Also, please include the names of highlighted transcripts in volcano plots (Figure 1A).

Response 4: Thank you for your constructive comment. As the reviewer suggested, we have revised the corresponding statement in the manuscript line 190. And we have annotated the top 5 dysregulated transcripts in Figure 1A (see the revised figure). We are sorry that considering labeling all the dysregulated transcripts in the volcano plots would make the labels too small to see, we just highlighted the representative transcripts.

Point 5: Line 190-191. The term “significantly changed”, is it different from differential expression?

Response 5: Thank you for your careful check. In line 190-191, what we want to express is that the 536 mRNA transcripts were significantly differentially expressed mRNAs based on FPKMs value. We have revised the corresponding statement in the manuscript.

Point 6: Figure 1E is missing. Please recheck the labels and legends.

Response 6: Thank you for your careful check. Considering the Figure 1D & 1E share a single legend, we have combined the Figure 1D & 1E as Figure 1D (see the revised figure).

Point 7: In Figure 2A legends please replace DEG with DEncRNA.

Response 7: Thank you for your comment, we have revised that.

Point 8: Section 3.4 (Line 226): “Based on the differential expression analysis in our initial microarray data “. Which microarray data authors trying to mention here. I believe the dataset in the previous section is based on the sequencing platform.

Response 8: Thank you for your comment. We are sorry for making possibly misguided statement. We have revised the statement in Line 231 as “Based on the differentially expression analysis in our sequencing data”.

Point 9: Please include WGCNA details in the method section.

Response 9: We are very grateful to your constructive comment. We have added relevant statement in the revised manuscript line 131-132.

Point 10: Please include a limitation section and mention the possible variables that can change the outcome, e.g. sample size.

Response 10: Thank you for your suggestion. We have added relevant statement as suggested in the revised manuscript discussion section, Line 454-457.

Reviewer 3 Report

In this study, ZHENG et al. focus on the relationships between Playtocodin treatment and deregulated coding/noncoding RNAs expression in lung cancer cell line A293. They describe a series of interesting results.

Authors performed whole transcriptome profiling (RNA sequencing, circRNA sequencing, and IncRNA sequencing) to investigate the RNA regulatory networks in the PD treated NSCLC cells versus non treated cells, using clear, well-defined and well-desribed volcanoplots, heatmaps, chord diagram, circos plots. Additionally, GSEA analysis and alternative splicing events analysis was elegantly described.

Besides, the molecular mechanism of PD action through network pharmacology and molecular docking technology emphasize impact of this manuscript.

On top of that, Authors performed and confirmed PD function and its candidate molecular targets by in vitro experiments.

Specifically, the relevance of PD in Lung cancer was rated as very good. The quality of writing was good. The technical merit was very good

correction in text:

Line 160 : 24h intead of 24

Author Response

We are very grateful for your careful check and giving us the positive comments. We have thoroughly revised the syntax error of the manuscript. 

correction in text:

Line 160 : 24h intead of 24

Response: Thank you for pointing this out, we have revised it.